# Reducing non-communicable diseases among Palestinian populations in Gaza: A participatory comparative and cost-effectiveness modeling assessment

**Sanjay Basu**[1,2]*, **John S. Yudkin**[3], **Mohammed Jawad**[4], **Hala Ghattas**[5,6], **Bassam Abu Hamad**[7], **Zeina Jamaluddine**[8], **Gloria Safadi**[5], **Marie-Elizabeth Ragi**[5], **Raeda El Sayed Ahmad**[5], **Eszter P. Vamos**[4], **Christopher Millett**[4,9]

1 Center for Vulnerable Populations, University of California San Francisco, San Francisco, California, United States of America, 2 Institute of Health Policy, Management & Evaluation, University of Toronto, Toronto, Canada, 3 Division of Medicine, Institute of Cardiovascular Science, University College London, London, United Kingdom, 4 Public Health Policy Evaluation Unit, School of Public Health, Imperial College London, London, United Kingdom, 5 Center for Research on Population and Health, American University of Beirut, Beirut, Lebanon, 6 Arnold School of Public Health, University of South Carolina, Columbia, South Carolina, United States of America, 7 School of Public Health, Al-Quds University, Jerusalem, Israel, 8 Faculty of Epidemiology and Population Health, London School of Hygiene & Tropical Medicine, London, United Kingdom, 9 National School of Public Health, Public Health Research Centre, Comprehensive Health Research Center, NOVA University Lisbon, Lisbon, Portugal

* sanjay.basu@ucsf.edu

**Data Availability Statement:** Data associated with this paper are available without restriction at https://github.com/sanjaybasu/ncdgaza.

## Abstract

We sought to assess the effectiveness and cost-effectiveness of potential new public health and healthcare NCD risk reduction efforts among Palestinians in Gaza. We created a micro-simulation model using: (i) a cross-sectional household survey of NCD risk factors among 4,576 Palestinian adults aged ≥40 years old in Gaza; (ii) a modified Delphi process among local public health experts to identify potentially feasible new interventions; and (iii) reviews of intervention cost and effectiveness, modified to the Gazan and refugee contexts. The survey revealed 28.6% tobacco smoking, a 40.4% prevalence of hypertension diagnosis (with a 95.6% medication treatment rate), a 25.6% prevalence of diabetes diagnosis (with 95.3% on treatment), a 21.9% prevalence of dyslipidemia (with 79.6% on a statin), and a 9.8% prevalence of asthma or chronic obstructive pulmonary disease (without known treatment). A calibrated model estimated a loss of 9,516 DALYs per 10,000 population over the 10-year policy horizon. The interventions having an incremental cost-effectiveness ratio (ICER) less than three times the GDP per capita of Palestine per DALY averted (<$10,992 per DALY averted)(<$10,992 per DALY averted) included bans on tobacco smoking in indoor and public places [$34 per incremental DALY averted (95% CI: $17, $50)], treatment of asthma using low dose inhaled beclometasone and short-acting beta-agonists [$140 per DALY averted (95% CI: $77, $207)], treatment of breast cancer stages I and II [$730 per DALY averted (95% CI: $372, $1,100)], implementing a mass media campaign for healthier nutrition [$737 per DALY averted (95% CI: $403, $1,100)], treatment of colorectal cancer stages I and II [$7,657 per DALY averted (95% CI: $3,721, $11,639)], and (screening with mammography [$17,054 per DALY averted (95% CI: $8,693, $25,359)]). Despite high levels of

**Funding:** This study was jointly funded by the UK's Department for International Development (DFID), the Medical Research Council (MRC), the Economic and Social Research Council (ESRC) and Wellcome Trust's Health Systems Research Initiative (HSRI) (MR/S012877/1). The funders played no role in the study design, results interpretation or the decision to submit for publication.

**Competing interests:** The authors have declared that no competing interests exist.

NCD risk factors among Palestinians in Gaza, we estimated that several interventions would be expected to reduce the loss of DALYs within common cost-effectiveness thresholds.

## Introduction

In recent years, the global health community has focused increasing attention on non-communicable disease (NCD) prevention and control. The World Health Organization (WHO) developed a Global Action Plan that aims to reduce premature NCD deaths by 25% by 2025 [1]. A key question for global humanitarian agencies is how to select among these interventions for resource-constrained settings like the Gaza Strip, an area that has faced ongoing conflict, blockade, and economic hardship for over a decade [2].

The Gaza Strip is a densely populated area of about 2 million Palestinian refugees and non-refugees [3]. About two-thirds of the population are refugees and rely on the United Nations Relief and Works Agency (UNRWA) and other aid for basic services. The health system in Gaza is fragmented, with care provided by public, private, and humanitarian groups like UNRWA, which operates 22 primary care centers [4]. However, UNRWA funding cuts threaten healthcare for the refugee population [5]. Several political and structural factors including Israeli policies of enforcing fishing limits, limitations and restrictions on food aid packages, agricultural land use limits, and the restriction of cancer treatment availability have also been documented as impacting on NCDs in Gaza [6–9].

While NCD prevalence data in Gaza are limited, available evidence suggests increasing rates of obesity, diabetes, hypertension, and other risk factors, especially among older adults. For example, 17.4% of those over 40 have hypertension and 11.8% have diabetes [10]. Tobacco use is also common, with 9% of youth and 40% of men using some form of tobacco [10]. Over the last several years, major strides have been made by UNRWA and other healthcare authorities to treat blood pressure and atherosclerotic cardiovascular disease risk, such that rates of hypertension treatment and control have increased dramatically, as has the use of statin therapy [11]. These CVD secondary prevention interventions are considered highly effective and cost-effective [12]. Other NCD risk-reduction interventions–particularly those directed at the societal level (such as used to improve nutrition, physical activity, and tobacco use), and for non-CVD NCDs (particularly cancers and lung diseases)—have lagged in planning and implementation in Gaza.

In this study, utilizing data from a newly available cross-sectional household survey of both refugee and non-refugee populations, we created a microsimulation model to assist in comparing the effectiveness and cost-effectiveness of potential new public health and healthcare NCD risk reduction efforts in Gaza, to follow the efforts of the last few years that focused primarily on medication treatments for CVD. We identified which interventions to compare through a modified Delphi process among a panel of local experts who identified the subset of WHO-recommended interventions that could feasibly be delivered in the Gazan setting. We then compared the effectiveness and cost-effectiveness of the interventions under different budget ceilings, based on a household cross-sectional health and nutrition survey, local cost data, and reviews of intervention effectiveness. We sought to provide insights into the interventions for NCD prevention and control within the resource-constrained Gazan context for consideration by UNRWA and related policymakers.

## Methods

We created and applied a microsimulation model using three sources of data: (i) a cross-sectional household survey providing information about baseline characteristics and NCD risk factors among the Palestinian population in Gaza; (ii) a modified Delphi process for local public health experts to identify potentially feasible interventions for modeling NCD risk reduction strategies in Gaza, including associated costs; and (iii) reviews and meta-analyses of intervention effectiveness, with modifications for the Gazan and refugee contexts. We utilized the model to compare the effectiveness and cost-effectiveness of the studied interventions. The study was approved by Al-Quds University, the Imperial College Research Ethics Committee (20IC5733), the American University of Beirut Institutional Review Board, the Gaza Helsinki Committee (PHRC/HC/483/19), and the UNRWA research review board. Due to armed conflict and COVID-19, leading to academic department closures and study personnel migration, the study extended over the period 2020–2023, with dates of individual study components detailed below.

### Data sources

**Cross-sectional household survey.** We conducted an interviewer-administered, face-to-face household survey in Gaza among adults aged 40 years and older. A sampling frame was drawn from enumeration areas of the 2017 Population and Housing Census and used as the basis for a random multistage, stratified cluster sampling approach to produce representative data of the Gazan population [13]. We specifically selected enumeration areas (the primary sampling units) from each sampling stratum (North Gaza, Gaza, Dier Al Balah, Khan Yunis, and Rafah governorates) and calculated the sample size as 4,520 participants from 2,443 households to detect within rounding error the estimated prevalence of coronary artery disease (11.3%) [9], assuming a response rate of 90.0% and design effect of 1.5.

We applied survey tools based on established approaches to household surveys, adapted to the local context, translated into Palestinian Arabic dialect, back-translated into English to test validity, and pilot tested in a random subset of households. The survey tool included an individual questionnaire with modules on sociodemographics, NCD history, and tobacco use–including the WHO Study on global AGEing and adult health (SAGE) Survey on sociodemographics and NCD history (including self-reported history of diabetes, hypertension, hyperlipidemia, cardiovascular disease, respiratory disease, and cancer) [14, 15], and the Global Adult Tobacco Survey (including cigarette and waterpipe use) [16]—both of which have versions validated in Arabic. Height, weight, and three blood pressure readings were obtained among all sampled individuals using validated instruments per the WHO SAGE protocol. A final sample size of 4,576 individuals across 2,445 households was achieved, with the recruitment period and data collection occurring between March 18 and July 15, 2020. A subset of 1,938 patients also had laboratory blood test results available from UNRWA clinics (specifically, creatinine, lipid panel and hemoglobin A1c) which were linked to the survey by individual participant unique identifiers and accessed for research purposes on July 14, 2020. Using anonymized identifiers, the authors did not have access to information that could identify individual participants during or after data collection.

**Modified Delphi process.** We conducted a modified Delphi process with a panel of 34 local public health experts in Gaza to review the WHO list of 85 recommended interventions for NCDs and determine which are most feasible in the Gaza context [17, 18]. The panel included representatives from the Ministry of Health, local universities, and non-governmental organizations with experience in NCD prevention and control in Gaza.

First, panelists were provided with the list of WHO NCD interventions and asked to score each intervention on a scale of 1 to 9 for 'feasibility' in Gaza, where 1 is 'not feasible' and 9 is 'highly feasible.' Panelists were also asked open-ended questions about challenges, costs, and implementation logistics associated with each intervention [18]. Responses were analyzed and summarized by the research team.

Next, panelists received a summary report with the first round of results and were asked to re-score each intervention after reviewing the input from the other panelists. They were also asked to highlight interventions they consider highest priorities for cost-effectiveness modeling.

The research team reviewed the results from two rounds of discussion to determine which interventions emerged as most feasible, taking into account both the quantitative scores and qualitative input on feasibility and prioritization. The research team extracted from the panelists' open-ended responses a list of factors that would need to be incorporated into cost-effectiveness modeling for the priority interventions. These include costs related to program delivery, health system and infrastructure logistics, challenges in changing behaviors and social norms, and indirect costs of the interventions [19]. The Delphi process was conducted in July and September 2021.

**Modeling methods.**  Modeling followed the 2022 Consolidated Health Economic Evaluation Reporting Standards (CHEERS, **S1 Table**) [20]. A microsimulation model was constructed, which is a population-representative, individual-level model that simulated the lifecourse of each person across ages 40 years and older in the Palestinian population of Gaza and computed their disability-adjusted life-years (DALYs) lost to NCDs without and with the selected interventions, and the costs of the interventions, over a 10-year policy time horizon from a societal perspective [9, 20, 21]. The incremental cost-effectiveness in 2023 international dollars per DALY averted was computed as each individual intervention was compared to the current status-quo level of exposure to that intervention per the survey, at a 3% annual discount rate for both dollars and DALYs.

We estimated the individual risk of each of the five major NCDs in the Palestinian population in Gaza (cardiovascular disease consisting of coronary heart disease or stroke, type 2 diabetes mellitus, asthma/COPD, breast cancer, and colorectal cancer) by using common risk scores derived from individual risk factors and validated in multi-ethnic populations including Middle Eastern or Arabic-speaking populations, calibrated to the estimated incidence of each disease by age and sex in the Palestinian population in Gaza [22]. For cardiovascular disease, we used the Globorisk score (averaging the laboratory-based risk scores from Jordan, Lebanon and Syria given the absence of a Palestine-specific model) [23]; for diabetes, the Finnish Diabetes Risk Score (FINDRISC) validated among Middle-Eastern populations [24, 25]; for asthma/COPD, a multivariate risk model with self-reported history of asthma/COPD and history of exposure to smoke as variables for the relative risk of morbidity or mortality from asthma or COPD with age [26–29]; for breast cancer, the Gail Breast Cancer Risk Assessment Tool (BCRT) validated among Middle-Eastern populations [30–32]; and for colorectal cancer, the ColoRectal Cancer Predicted Risk Online (CRC-PRO) risk calculator [33]. Calibration to the Palestinian population in Gaza was performed by computing each risk score across each individual in the population-representative study sample, then scaling the risk scores to achieve the estimated incidence of each NCD in the Palestinian population from the Global Burden of Disease project [22]. For risk scores, missing data were imputed using multiple imputation with chained equations, leveraging a classification and regression tree model [34].

The reduction in risk of each NCD based on each intervention was assessed by a review following the Preferred Reporting Items for Systematic Reviews and Meta-Analyses (PRISMA) guidelines [35]. For each intervention, we reviewed the available meta-analytic estimates of the

intervention's effect on disease incidence and mortality (see **S1 Text**). We started with the WHO's literature review that led to the WHO's recommendation of the intervention [1, 17], and expanded the review on PubMed.gov and Google Scholar, as well as using two artificial intelligence tools (elicit.org and consensus.app) that enabled finding of original research articles, reviews, and meta-analyses for each of the interventions.

We assessed costs related to each intervention. Direct costs were assessed using the WHO Choosing Interventions that are Cost-Effective (CHOICE) approach [19], which considers both individual-level and program-level costs. Individual-level costs included medications, diagnostics, health facility visits and associated materials and personnel expenditures per individual exposed to the intervention. Program-level costs included costs of administration, monitoring and evaluation, supervision and training. We used the WHO list of 14 publicly available datasets for non-traded cost variables (locally-produced or human resources) as well as traded items (purchased on the market) within Gaza, using data from UNRWA where available. Costs unavailable in Gaza were obtained from other Middle Eastern countries, then adjusted by the GDP purchasing power parity per capita between the other countries and Palestine [36].

We computed DALYs associated with each NCD by using the disability weights estimated from the Global Burden of Disease project for each NCD [37], and computing the incidence of the NCD in each simulated individual based on the risk of the NCD given their individual-level risk factors per the survey and associated risk score for each NCD, net of competing risks.

The model was programed in *R* in May through July of 2023, with code shared online alongside a pre-specified protocol at: https://github.com/sanjaybasu/ncdgaza

**Sensitivity and uncertainty analysis.** Sensitivity analysis was performed on the population reach of each intervention. Population reach was simulated by collecting information in the review on the range of practical achievement of access to prior interventions among refugee populations in Gaza, and varying the intervention benefits in the model to the associated subset of the population, starting from a 64% reach rate with a linear implementation period over the 10 year time horizon [38].

Uncertainty analysis was performed by repeatedly sampling with replacement 10,000 times from normal distributions constructed from the mean and standard deviation of each input parameter value and re-running the model with each sample to identify the mean and 95% credible interval around the incremental cost-effectiveness ratio for each intervention [39].

## Results

### Characteristics of the study population

The cross-sectional household survey produced data on 4,576 adults across 2,445 households. The surveyed population had a mean age of 57.1 years old (SD: 10.2, range: 40–80), of which 2,473 people (54.0%) identified as women, and 3,136 (68.5%) identified as refugees, paralleling the 2017 Census [13]. The self-reported prevalence of common NCDs and NCD risk factors (**Table 1**) included: 25.6% of people reporting a diagnosis of diabetes (n = 1,172), 40.4% of people reporting hypertension (n = 1,849), 21.9% reporting dyslipidemia (n = 1,001), and 9.8% reporting asthma or COPD (n = 449). Of those reporting a diagnosis of diabetes, 95.3% (n = 1,117) reported taking a prescribed medication for diabetes; among those with hypertension, 95.6% (n = 1,767) reported taking a prescribed medication for hypertension; among those with dyslipidemia, 79.6% (n = 796) reported taking a statin; but among those with asthma/COPD, none reported controller medication use.

Laboratory values and exam-based measurements are also shown in **Table 1**. Mean hemoglobin A1c was 12.4% (SD: 1.8%), with a mean of 12.5% (SD: 1.7%) among those reporting a

**Table 1. Characteristics of the study population of Palestinians in Gaza per household survey of ages 40+ years old (2020).**

| | Male | Female | Standardized mean difference |
|---|---|---|---|
| n | 2103 | 2473 | |
| Age, yrs, mean (SD) | 58.6 (10.2) | 55.4 (10.5) | 0.308 |
| Diabetes diagnosis, n (%) | 525 (25.0) | 647 (26.2) | 0.027 |
| Diabetes medications, n (%) | 498 (23.7) | 619 (25.0) | 0.031 |
| Hypertension diagnosis, n (%) | 798 (37.9) | 1051 (42.5) | 0.093 |
| Hypertension medications, n (%) | 761 (36.2) | 1006 (40.7) | 0.092 |
| Systolic blood pressure, mmHg (SD) | 132.2 (17.1) | 128.0 (18.7) | 0.235 |
| Dyslipidemia diagnosis, n (%) | 450 (21.4) | 551 (22.3) | 0.021 |
| Statin medication, n (%) | 492 (23.4) | 552 (22.3) | 0.026 |
| Asthma/COPD diagnosis, n (%) | 212 (10.1) | 237 (9.6) | 0.017 |
| Tobacco smoking, n (%) | 1280 (60.9) | 29 (1.2) | 1.689 |
| Water pipe smoking, n (%) | 304 (14.5) | 22 (0.9) | 0.527 |
| Body mass index, kg/m^2, mean (SD) | 28.9 (5.4) | 33.5 (6.4) | 0.779 |
| Subset with laboratory values | | | |
| n | 785 | 1153 | |
| Total cholesterol, mg/dL, mean (SD) | 170.5 (38.8) | 181.9 (40.5) | 0.287 |
| Triglycerides, mg/dL, mean (SD) | 178.8 (144.1) | 167.4 (121.5) | 0.086 |
| Low-density lipoprotein cholesterol, mg/dL, mean (SD) | 106.3 (37.7) | 112.9 (39.2) | 0.171 |
| High-density lipoprotein cholesterol, mg/dL, mean (SD) | 41.5 (14.0) | 46.5 (16.5) | 0.326 |
| Serum creatinine, mg/dL (SD) | 1.0 (0.5) | 0.8 (0.4) | 0.439 |
| Hemoglobin A1c, mean % (SD), among those with diabetes | 13.3 (1.5) | 11.5 (1.4) | 1.201 |

previous diabetes diagnosis and 12.5% (SD: 1.7%) among those reporting both a diagnosis and taking diabetes medications. Mean systolic blood pressure in the sample was 130.0 mmHg (SD: 18.1 mmHg), with a mean of 136.3 mmHg (SD: 18.4 mmHg) among those reporting a diagnosis of hypertension and 136.1 mmHg (SD: 18.3 mmHg) among those reporting both a hypertension diagnosis and hypertension medication use. Mean low-density lipoprotein (LDL) cholesterol was 107.5 mg/dL (SD: 38.1 mg/dL), with a mean of 111.6 mg/dL (SD: 40.1 mg/dL) among those reporting a dyslipidemia diagnosis, and 111.8 mg/dL (SD: 40.4 mg/dL) among those reporting both a dyslipidemia diagnosis and treatment with a statin. The prevalence of tobacco smoking was 28.6% (n = 1,309, mostly among men with 1,280 males smoking; 60.9%), and water pipe smoking was 7.1% (n = 329, with 304 being male). Mean body mass index was 31.4 kg/m^2 (SD: 6.4 kg/m^2).

## Interventions selected by Delphi process

The Delphi process among the 85 WHO-recommended NCD interventions resulted in the selection of 12 interventions, of which nine were judged to be feasible for modeling (**Table 2**). The modeling-amenable interventions were in the WHO NCD intervention domains of reducing tobacco use; unhealthy diet; physical inactivity; managing cancer; and managing chronic respiratory disease.

## Model-based comparative and cost-effectiveness estimates

The calibrated model estimated a loss of 9,516 (95% CI: 7,947, 11,201) DALYs per 10,000 population over the 10-year policy horizon. Among the NCDs simulated, the greatest loss of DALYs was from CVD (3,718 DALYs lost per 10,000; 95% CI: 3,299, 4,174), followed by

**Table 2. Interventions selected via the modified Delphi process among local experts on public health in Gaza, 2021.**

| World Health Organization NCD category | Intervention number and description per World Health Organization NCD Intervention List, selected by modified Delphi process | Considered feasible for modeling cost-effectiveness in Gaza, per modeling team? |
|---|---|---|
| Reduce tobacco use | 1.4 Eliminate exposure to second-hand tobacco smoke in all indoor workplaces, public places, public transport | Yes |
| Reduce unhealthy diet | 2.11 Implement nutrition education and counseling in different settings (for example, in preschools, schools, workplaces and hospitals) to increase the intake of fruits and vegetables | Yes |
| | 2.13 Implement mass media campaign on healthy diets, including social marketing to reduce the intake of total fat, saturated fats, sugars and salt, and promote the intake of fruits and vegetables | Yes |
| Reduce physical inactivity | 3.1 Implement community wide public education and awareness campaign for physical activity which includes a mass media campaign combined with other community based education, motivational and environmental programmes aimed at supporting behavioral change of physical activity levels | Yes |
| | 3.2 Provide physical activity counseling and referral as part of routine primary health care services through the use of a brief intervention | Yes |
| | 3.4 Implement whole-of-school programme that includes quality physical education, availability of adequate facilities and programs to support physical activity for all children | No, insufficient data on school facilities in Gaza |
| Manage cancer | 6.2 Screening with mammography (once every 2 years for women aged 50–69 years) linked with timely diagnosis and treatment of breast cancer | Yes |
| | 6.3 Treatment of colorectal cancer stages I and II with surgery +/- chemotherapy and radiotherapy | Yes |
| | 6.4 Treatment of breast cancer stages I and II with surgery +/- systemic therapy | Yes |
| Manage chronic respiratory disease | 7.3 Treatment of asthma using low dose inhaled beclometasone and short acting beta agonist | Yes |
| Enhancing psychosocial and mental health status | 8.2 Screening and identification of risky cases | No, insufficient survey data on risk and intervention impact of screening |
| Enforcing governance of services for adequate prevention, screening, and management of NCDs (like better regulatory measures, increasing coordination and fiscal measures) | 9.1.5 Using standardized protocols and guidelines for screening, diagnosis and management | No, insufficient data on impact of standardization |

diabetes (2,088 DALYs lost per 10,000; 95% CI: 1,787, 2,402), then asthma/COPD (1,856 DALYs lost per 10,000; 95% CI: 1,465, 2,270; Table 3).

For ease of reading, we have ordered the presentation of results from interventions with the lowest incremental cost per DALY averted to the highest. Fig 1A and Fig 1B display the incremental cost and DALYs averted for each of the simulated interventions over the 10-year time horizon, per 10,000 Palestinian people aged 40+ years in Gaza.

## Eliminate exposure to second-hand tobacco smoke in all indoor workplaces, public places, and public transport

The simulation of eliminating exposure to second-hand tobacco smoke in indoor workspaces, public places and public transport reduced the loss of DALYs by 433 (95% CI: 429, 445; Table 3), primarily from reduced CVD (404 DALYs, 95% CI: 403, 431), then reduced asthma and COPD (29, 95% CI: 16, 29). At a mean cost of $0.20 per person per year, the incremental cost-effectiveness ratio (ICER) for the intervention was $34 per incremental DALY averted (95% CI: $17, $50).

**Table 3. Simulation modeling results of the incidence, mortality and disability-adjusted life-years (DALYs) associated with several non-communicable diseases among the Palestinian population of Gaza over a 10-year time horizon among 10,000 adults aged 40 years and older, 2023.**

| Condition | Total | | | Change from baseline | | |
|---|---|---|---|---|---|---|
| | Incidence | Mortality | DALYs | Incidence | Mortality | DALYs |
| Baseline | | | | | | |
| CVD | 3417 (3334, 3512) | 163 (140, 190) | 3718 (3299, 4174) | NA | NA | NA |
| Diabetes | 1393 (1327, 1463) | 294 (258, 331) | 2088 (1787, 2402) | NA | NA | NA |
| COPD/Asthma | 699 (651, 746) | 86 (69, 105) | 1856 (1465, 2270) | NA | NA | NA |
| Breast Cancer | 90 (74, 110) | 48 (35, 63) | 900 (639, 1190) | NA | NA | NA |
| Colorectal Cancer | 114 (94, 136) | 84 (67, 102) | 954 (757, 1165) | NA | NA | NA |
| Total | 5713 (5480, 5967) | 675 (569, 791) | 9516 (7947, 11201) | NA | NA | NA |
| 1.4 Eliminate exposure to second-hand tobacco smoke in all indoor workplaces, public places, public transport | | | | | | |
| CVD | 3114 (3027, 3202) | 149 (125, 173) | 3314 (2896, 3743) | -303 (-307, -310) | -14 (-15, -17) | -404 (-403, -431) |
| Diabetes | 1393 (1327, 1463) | 294 (258, 331) | 2088 (1787, 2403) | 0 (0, 0) | 0 (0, 0) | 0 (0, 0) |
| COPD/Asthma | 688 (638, 736) | 85 (68, 103) | 1827 (1439, 2254) | -11 (-13, -10) | -1 (-1, -2) | -29 (-26, -16) |
| Breast Cancer | 90 (74, 110) | 48 (35, 63) | 900 (639, 1191) | 0 (0, 0) | 0 (0, 0) | 0 (0, 0) |
| Colorectal Cancer | 114 (94, 136) | 84 (67, 102) | 954 (757, 1165) | 0 (0, 0) | 0 (0, 0) | 0 (0, 0) |
| Total | 5399 (5160, 5647) | 660 (553, 772) | 9083 (7518, 10756) | -314 (-320, -320) | -15 (-16, -19) | -433 (-429, -445) |
| 2.11 Implement nutrition education and counseling in different settings (for example, in preschools, schools, workplaces and hospitals) to increase the intake of fruits and vegetables | | | | | | |
| CVD | 3408 (3325, 3500) | 163 (138, 189) | 3706 (3295, 4159) | -9 (-9, -12) | -1 (-2, 0) | -12 (-4, -15) |
| Diabetes | 1390 (1326, 1460) | 294 (258, 331) | 2084 (1781, 2399) | -3 (-1, -3) | 0 (0, 0) | -4 (-6, -3) |
| COPD/Asthma | 699 (651, 746) | 86 (70, 105) | 1856 (1466, 2270) | 0 (0, 0) | 0 (0, 0) | 0 (0, 0) |
| Breast Cancer | 90 (74, 110) | 48 (35, 63) | 898 (639, 1188) | 0 (0, 0) | 0 (0, 0) | 0 (0, 0) |
| Colorectal Cancer | 114 (94, 136) | 84 (67, 102) | 953 (756, 1165) | 0 (0, 0) | 0 (0, 0) | 0 (0, 0) |
| Total | 5701 (5470, 5952) | 675 (568, 790) | 9497 (7937, 11181) | -12 (-10, -15) | -1 (-2, 0) | -16 (-10, -20) |
| 2.13 Implement mass media campaign on healthy diets, including social marketing to reduce the intake of total fat, saturated fats, sugars and salt, and promote the intake of fruits and vegetables | | | | | | |
| CVD | 3403 (3320, 3495) | 163 (138, 189) | 3699 (3289, 4154) | -14 (-14, -17) | -1 (-2, 0) | -19 (-10, -20) |
| Diabetes | 1389 (1324, 1458) | 293 (257, 331) | 2081 (1779, 2397) | -4 (-3, -5) | -1 (-1, 0) | -5 (-8, -5) |
| COPD/Asthma | 699 (651, 746) | 86 (69, 105) | 1856 (1466, 2270) | 0 (0, 0) | 0 (0, 0) | 0 (0, 0) |
| Breast Cancer | 90 (74, 110) | 48 (35, 63) | 898 (639, 1188) | 0 (0, 0) | 0 (0, 0) | -1 (-2, 0) |
| Colorectal Cancer | 114 (94, 135) | 84 (67, 102) | 952 (756, 1165) | 0 (0, -1) | 0 (0, 0) | -1 (-2, 0) |
| Total | 5695 (5463, 5944) | 674 (566, 790) | 9486 (7929, 11174) | -18 (-17, -23) | -2 (-3, -1) | -26 (-18, -27) |
| 3.1 Implement community wide public education and awareness campaign for physical activity which includes a mass media campaign combined with other community based education, motivational and environmental programmes aimed at supporting behavioral change of physical activity levels | | | | | | |
| CVD | 3406 (3324, 3499) | 163 (138, 189) | 3703 (3294, 4156) | -11 (-10, -13) | -1 (-2, 0) | -15 (-5, -18) |
| Diabetes | 1388 (1323, 1456) | 293 (256, 331) | 2079 (1778, 2397) | -5 (-4, -7) | -1 (-2, 0) | -9 (-9, -5) |
| COPD/Asthma | 696 (648, 744) | 86 (68, 105) | 1849 (1464, 2264) | -3 (-3, -2) | 0 (-1, 0) | -7 (-1, -13) |
| Breast Cancer | 90 (74, 110) | 48 (35, 63) | 897 (639, 1184) | 0 (0, 0) | 0 (0, 0) | -3 (0, -6) |
| Colorectal Cancer | 113 (93, 135) | 84 (66, 102) | 951 (750, 1165) | -1 (-1, -1) | 0 (-1, 0) | -3 (-7, 0) |
| Total | 5693 (5462, 5944) | 674 (563, 790) | 9479 (7925, 11166) | -20 (-18, -23) | -1 (-6, -1) | -37 (-22, -42) |
| 3.2 Provide physical activity counseling and referral as part of routine primary health care services through the use of a brief intervention | | | | | | |
| CVD | 3389 (3304, 3475) | 162 (137, 188) | 3680 (3245, 4131) | -28 (-30, -27) | -1 (-3, -2) | -38 (-54, -33) |
| Diabetes | 1378 (1314, 1446) | 291 (255, 329) | 2064 (1757, 2390) | -15 (-13, -17) | -3 (-3, -2) | -24 (-30, -12) |
| COPD/Asthma | 692 (643, 740) | 85 (68, 104) | 1838 (1456, 2262) | -7 (-8, -6) | -1 (-1, -1) | -18 (-9, -27) |
| Breast Cancer | 90 (73, 109) | 47 (34, 63) | 894 (637, 1183) | -1 (-2, 0) | -1 (-1, 0) | -6 (-2, -7) |
| Colorectal Cancer | 113 (93, 134) | 83 (66, 101) | 946 (745, 1161) | -1 (-1, -2) | -1 (-1, -1) | -8 (-12, -4) |
| Total | 5662 (5427, 5904) | 668 (560, 785) | 9422 (7840, 11127) | -51 (-53, -63) | -7 (-9, -6) | -94 (-107, -93) |
| 6.2 Screening with mammography (once every 2 years for women aged 50–69 years) | | | | | | |

*(Continued)*

**Table 3.** (Continued)

| Condition | Total | | | Change from baseline | | |
|---|---|---|---|---|---|---|
| | Incidence | Mortality | DALYs | Incidence | Mortality | DALYs |
| CVD | 3417 (3334, 3512) | 163 (140, 190) | 3718 (3299, 4175) | 0 (0, 0) | 0 (0, 0) | 0 (0, 0) |
| Diabetes | 1393 (1327, 1463) | 294 (258, 331) | 2088 (1787, 2403) | 0 (0, 0) | 0 (0, 0) | 0 (0, 0) |
| COPD/Asthma | 699 (651, 746) | 86 (69, 105) | 1856 (1466, 2270) | 0 (0, 0) | 0 (0, 0) | 0 (0, 0) |
| Breast Cancer | 90 (74, 110) | 43 (31, 56) | 813 (580, 1063) | 0 (0, 0) | -5 (-4, -7) | -87 (-59, -127) |
| Colorectal Cancer | 114 (94, 136) | 84 (67, 102) | 954 (757, 1165) | 0 (0, 0) | 0 (0, 0) | 0 (0, 0) |
| Total | 5713 (5480, 5967) | 670 (565, 784) | 9429 (7889, 11076) | 0 (0, 0) | -5 (-4, -7) | -87 (-58, -125) |
| 6.3 Treatment of colorectal cancer stages I and II with surgery +/- chemotherapy and radiotherapy | | | | | | |
| CVD | 3417 (3334, 3512) | 163 (140, 190) | 3718 (3300, 4175) | 0 (0, 0) | 0 (0, 0) | 0 (0, 0) |
| Diabetes | 1393 (1327, 1463) | 294 (258, 331) | 2088 (1787, 2403) | 0 (0, 0) | 0 (0, 0) | 0 (0, 0) |
| COPD/Asthma | 699 (651, 746) | 86 (69, 105) | 1856 (1466, 2270) | 0 (0, 0) | 0 (0, 0) | 0 (0, 0) |
| Breast Cancer | 90 (74, 110) | 47 (35, 63) | 900 (639, 1191) | 0 (0, 0) | 0 (0, 0) | 0 (0, 0) |
| Colorectal Cancer | 114 (94, 136) | 70 (54, 87) | 794 (609, 988) | 0 (0, 0) | -14 (-13, -15) | -160 (-148, -177) |
| Total | 5713 (5480, 5967) | 660 (556, 776) | 9356 (7801, 11027) | 0 (0, 0) | -15 (-13, -15) | -160 (-146, -174) |
| 6.4 Treatment of breast cancer stages I and II with surgery +/- systemic therapy | | | | | | |
| CVD | 3417 (3334, 3512) | 163 (140, 190) | 3718 (3300, 4175) | 0 (0, 0) | 0 (0, 0) | 0 (0, 0) |
| Diabetes | 1393 (1327, 1463) | 294 (258, 331) | 2088 (1787, 2403) | 0 (0, 0) | 0 (0, 0) | 0 (0, 0) |
| COPD/Asthma | 699 (651, 746) | 86 (69, 105) | 1856 (1466, 2270) | 0 (0, 0) | 0 (0, 0) | 0 (0, 0) |
| Breast Cancer | 90 (74, 110) | 36 (25, 49) | 683 (459, 932) | 0 (0, 0) | -12 (-10, -14) | -217 (-180, -258) |
| Colorectal Cancer | 114 (94, 136) | 84 (67, 102) | 954 (757, 1165) | 0 (0, 0) | 0 (0, 0) | 0 (0, 0) |
| Total | 5713 (5480, 5967) | 663 (559, 777) | 9299 (7769, 10945) | 0 (0, 0) | -12 (-10, -14) | -217 (-178, -256) |
| 7.3 Treatment of asthma using low dose inhaled beclometasone and short acting beta agonist | | | | | | |
| CVD | 3417 (3334, 3512) | 163 (140, 190) | 3718 (3300, 4175) | 0 (0, 0) | 0 (0, 0) | 0 (0, 0) |
| Diabetes | 1393 (1327, 1463) | 294 (258, 331) | 2088 (1787, 2403) | 0 (0, 0) | 0 (0, 0) | 0 (0, 0) |
| COPD/Asthma | 666 (615, 715) | 82 (64, 100) | 1766 (1380, 2173) | -33 (-36, -31) | -4 (-5, -3) | -90 (-85, -97) |
| Breast Cancer | 90 (74, 110) | 48 (35, 63) | 900 (639, 1191) | 0 (0, 0) | 0 (0, 0) | 0 (0, 0) |
| Colorectal Cancer | 114 (94, 136) | 84 (67, 102) | 954 (757, 1165) | 0 (0, 0) | 0 (0, 0) | 0 (0, 0) |
| Total | 5680 (5444, 5936) | 671 (564, 786) | 9426 (7863, 11107) | -33 (-36, -31) | -4 (-5, -3) | -90 (-84, -94) |

cardiovascular disease (CVD), chronic obstructive pulmonary disease (COPD).

### Treatment of asthma using low dose inhaled beclometasone and short acting beta agonist

Treatment of asthma using low dose inhaled beclometasone and short acting beta agonist reduced the loss of DALYs by 90 (95% CI: 84, 94) entirely through reduced asthma/COPD exacerbation incidence and mortality. At a mean cost of $24 per person per year with asthma or COPD, the ICER for the intervention was $140 per DALY averted (95% CI: $77, $207).

**Treatment of breast cancer stages I and II with surgery +/- systemic therapy.** Treatment of breast cancer stages I and II reduced the loss of DALYs by 217 (95% CI: 178, 256) entirely through reduced breast cancer mortality. At a mean cost of $1,423 per person with incident breast cancer stages I or II, the ICER for the intervention was $730 per DALY averted (95% CI: $372, $1,100).

**Implement a mass media campaign on healthy diets, including social marketing to reduce the intake of total fat, saturated fats, sugars and salt, and promote the intake of fruits and vegetables.** The simulation of implementing a mass media campaign on healthy diets reduced the loss of DALYs by 30 (95% CI: 18, 37), primarily through reduced CVD (19

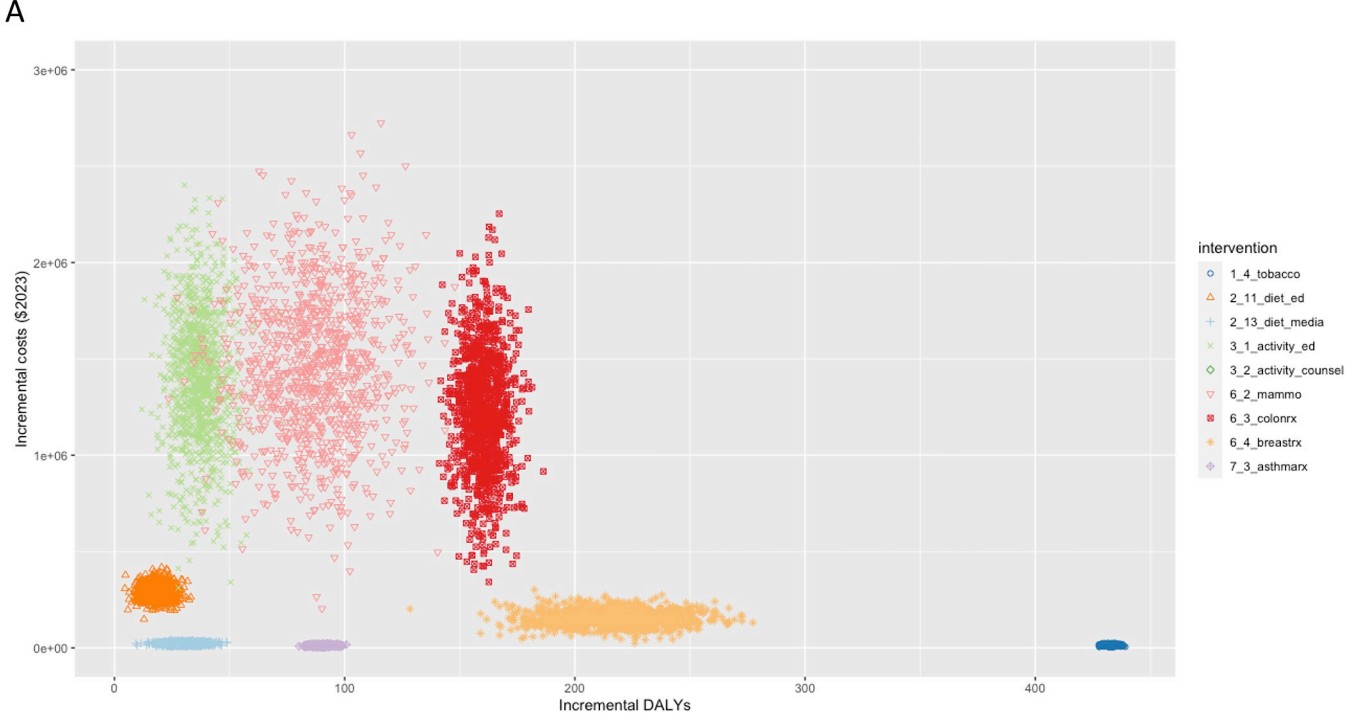

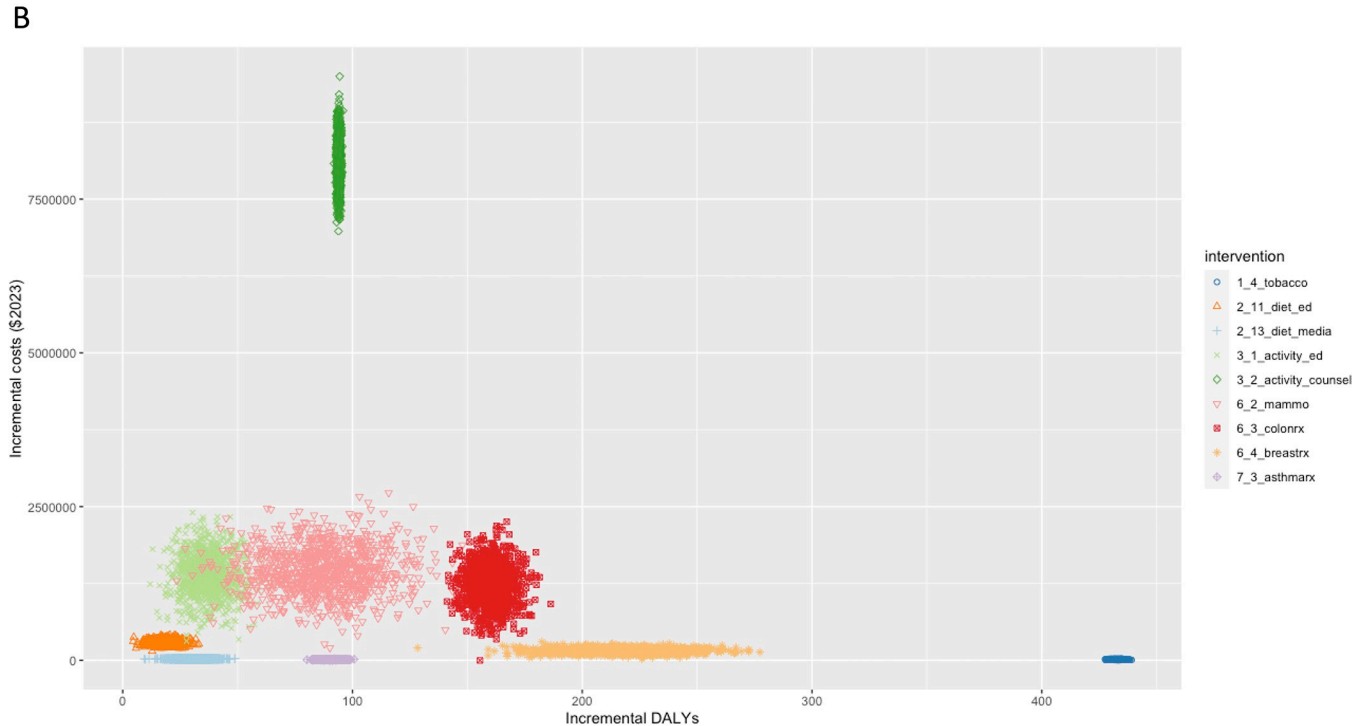

**Fig 1. Cost-effectiveness plane for selected non-communicable disease interventions among the Palestinian population of Gaza over a 10-year time horizon among 10,000 adults aged 40 years and older, 2023.** Incremental disability-adjusted life-years (DALYs) and incremental costs are presented at a 3% annual discount rate, with dollars expressed in 2023 International dollars. Cardiovascular disease (CVD). Intervention labels correspond to **Table 2**. (A) Across all interventions; (B) Zooming in on the lower left quadrant of the cost-effectiveness plane.

DALYs, 95% CI: 10, 20), followed by reduced diabetes (7 DALYs, 95% CI: 5, 8). At a mean cost of $0.30 per person per year for the intervention, the ICER for the intervention was $737 per DALY averted (95% CI: $403, $1,100).

**Treatment of colorectal cancer stages I and II with surgery +/- chemotherapy and radiotherapy.** Treatment of colorectal cancer stages I and II reduced the loss of DALYs by 160 (95% CI: 146, 174) entirely through reduced colorectal cancer mortality. At a mean cost of $14,736 per person with incident colorectal cancer stages I or II, the ICER for the intervention was $7,657 per DALY averted (95% CI: $3,721, $11,639).

**Implement nutrition education and counseling in different settings (for example, in preschools, schools, workplaces and hospitals) to increase the intake of fruits and vegetables.** The simulation of implementing nutrition education and counseling in schools, workplaces and hospitals to increase the intake of fruits and vegetables reduced the loss of DALYs by 19 (95% CI: 10, 20), primarily through reduced CVD (12 DALYs, 95% CI: 4, 15), then diabetes (3, 95% CI: 1, 3). At a mean cost of $4 per person per year, the ICER for the intervention was $14,840 per incremental DALY averted (95% CI: $11,343, $18,115).

**Screening with mammography (once every 2 years for women aged 50–69 years).** Screening with mammography reduced the loss of DALYs by 87 (95% CI: 48, 125) entirely through reduced breast cancer incidence and mortality. At a mean cost of $20 per person per year (after accounting for the every-two-year rate and the proportion of the population who are female), the ICER for the intervention was $17,054 per DALY averted (95% CI: $8,693, $25,359).

**Implement community wide public education and awareness campaign for physical activity which includes a mass media campaign combined with other community based education, motivational and environmental programmes aimed at supporting behavioral change of physical activity levels.** Implementing a community wide public education and awareness campaign for physical activity reduced the loss of DALYs by 37 (95% CI: 22, 42), primarily through reduced CVD (15 DALYs, 95% CI: 5, 18), followed by diabetes (9 DALYs, 95% CI: 5, 9), asthma/COPD (7 DALYs, 95% CI: 1, 13), breast cancer (3 DALYs, 95% CI: 0, 6) and colorectal cancer (3 DALYs, 95% CI: 0, 7). At a mean cost of $19 per person per year, the ICER for the intervention was $38,025 per DALY averted (95% CI: $19,898, $55,680).

**Provide physical activity counseling and referral as part of routine primary health care services through the use of a brief intervention.** Providing physical activity counseling and referral as part of routine primary care reduced the loss of DALYs by 94 (95% CI: 93, 107), primarily through reduced CVD (38 DALYs, 95% CI: 43, 54), then diabetes (24 DALYs, 95% CI: 12, 30), COPD/asthma (18 DALYs, 95% CI: 9, 27), breast cancer (6 DALYs, 95% CI: 2, 7), and colorectal cancer (8 DALYS, 4, 12). At a cost of $110 per person per year, the ICER for the intervention was $86,023 (95% CI: $77,771, $93,740). The greatest uncertainty and range of costs (as shown in **Fig 1**) was for the primary care-based physical activity interventions, given a range of possible costs for training and labor for additional personnel to deliver the intervention, as discussed further in **S1 Text**.

## Sensitivity and uncertainty analyses

Sensitivity analyses varying the population reach of each intervention had a linear effect on DALYs and costs, not changing the incremental cost-effectiveness ratio among interventions. Uncertainty analyses are reflected in the 95% credible intervals around the mean results.

## Discussion

We utilized data from a cross-sectional household survey of both refugee and non-refugee populations in Gaza and found high levels of NCD risk factors and common diseases, along

with high treatment rates for hypertension, diabetes and dyslipidemia. Through a modified Delphi process, we further gathered from a panel of local experts that a set of further NCD interventions–related to population-level interventions to promote nutrition, physical activity and reduced tobacco use, alongside improved screening and treatment for asthma/COPD and cancers–were of key interest to further address NCDs. Based on a microsimulation model, we found that several WHO-recommended interventions of interest to local experts were expected to reduce the loss of DALYs from NCDs by between 30 and 433 DALYs, with a wide-ranging ICER from as low as $34 per DALY averted (for a indoor and public place tobacco ban, which had high impact but low cost to implement) to as high as $86,023 (for physical activity counseling in primary care, which had low impact despite moderate cost). The interventions having less than three times the GDP per capita of Palestine per DALY averted (<$10,992 per capita [36]) included indoor/public tobacco bans, nutrition education and counseling in schools and workspaces, mass media on healthy diets, treatment of breast or colorectal cancer stages I and II, and treatment of asthma/COPD. For reference, pharmaco-therapy-based services to treat hypertension, diabetes, and dyslipidemia per WHO guidelines have been previously found to have ICERs around $7,200 to $16,700 per DALY averted in low- and middle-income countries, per our prior assessment [40].

Our analysis helps to inform the second wave of NCD control efforts within UNRWA and other Gaza-focused public health and healthcare entities. The first wave of efforts focused on highly effective and cost-effective expansion of hypertension, dyslipidemia and diabetes treatments. Consistent with the first wave effort, the household study we studied revealed high levels of treatment among those diagnosed with hypertension, diabetes and dyslipidemia. Further primary prevention efforts in the public sphere and secondary prevention or treatment efforts for non-CVD NCDs have lagged behind. Our study reveals high interest in such efforts, as well as both potential effectiveness and cost-effectiveness of such interventions in the Gazan context. It is notable that Palestine (including both the West Bank and Gaza) is not mentioned in the recent tables compiling the NCD national capacity policies, strategies and action plans for the WHO, but our current assessment suggests that such prioritization may be feasible and important from a public health perspective [41, 42].

There are several limitations to our analysis. The largest limitation is that the analysis presented here was conducted prior to the October 2023 invasion of Gaza by Israeli forces, which profoundly changed the demographics and population health of the Gaza population. At the time of this writing in February 2024, more than 25,000 people have been killed according to the Gazan health ministry, and the United Nations has warned of a large-scale humanitarian crisis due to lack of food, housing, and clean water among other basic needs [43]. We cannot clearly anticipate what changes to communicable and non-communicable disease will take place in Gaza as a result of this escalation in conflict [44], but we can be certain that healthcare and public health infrastructure has been severely damaged and constrained, making our assessment of longer-term investments all the more important. Additionally, it is notable that the expert participants in our Delphi process identified that even prior to the recent conflict, there was insufficient data to assess the prevalence of different mental health needs among the population; no doubt the needs will have increased since the recent escalation in conflict. Much of our analysis also focused on advising UNRWA, yet since the conflict started, UNRWA has faced significant reductions in their ability to operate healthcare centers while simultaneously facing overwhelming demand, and it remains unclear whether and how UNRWA will be able to rebuild its infrastructure and capabilities in the future [45].

Beyond the immediate conflict, the cross-sectional survey used to understand the prevalence of disease was focused on adults over 40 years of age, yet it is known that as societal changes occur in Palestine, NCDs also appear among younger populations. Additionally,

laboratory data were only available for a subset of participants, subject to the selection of healthcare providers who had obtained laboratory measurements among those typically diagnosed with NCDs (e.g., hemoglobin A1c was primarily obtained among those diagnosed with diabetes). Our modeling efforts were based on meta-analytic data among those outside of Palestine, adjusted for the typical population reach of public health and healthcare interventions in Gaza. Nevertheless the effectiveness in Gaza would be expected to vary in the Gazan context based on how well such efforts may be culturally tailored and effectively disseminated to the population. Additionally, the costs of interventions were based on the closest-available cost estimates in the Middle East, adjusted for the purchasing power parity in Palestine. Further costs may be imposed on public health and healthcare entities operating in Palestine based on the impact on the supply chain of conflicts. Conversely, UNRWA and other international agencies have secured lower prices for some healthcare treatments given economies of scale in bulk purchasing. All of our modeling was based on risk models whose validity can vary among different ethnic subpopulations, and therefore our calibration to the incidence and mortality trends in Palestine may nevertheless not account for individual-level variations in risk. Finally, the impact of change in food assistance has many political and structural uncertainties as agricultural land in Gaza is very limited, and we have separately modeled the cost-effectiveness of changing the nutritional content of food aid packages to assist in nutrition improvement in the area, finding high uncertainty in the outcomes depending on approval processes for package content [9]. The potential cost-effectiveness of education, counseling and mass media on diets in Gaza, while considered feasible per the Delphi process, also had high uncertainty because of structural barriers that may play a role in being able to put these educational recommendations to practice, including availability and access of food.

## Conclusions

While effectiveness and cost-effectiveness remain uncertain in a conflict setting, our modeling efforts can help inform the relative scale of intervention impact and provide a sense of the degree of uncertainty when planning public health and healthcare interventions with best available data. Under the highly uncertain context of planning for health improvements among Palestinians in Gaza, our findings suggest a need for further expansion of NCD interventions, particularly in the planning of healthcare system re-building after conflict.

## Supporting information

**S1 Checklist. Inclusivity in global research.**
(DOCX)

**S1 Text.**
(DOCX)

**S1 Table. CHEERS checklist.**
(DOCX)

## Acknowledgments

We would like to thank the Palestinian Central Bureau of Statistics for their methodological support, UNRWA for sharing laboratory data, and the tremendous efforts of our data collectors and field supervisors in the Gaza Strip who showed courage, resilience, and dedication to conduct this survey in the most challenging of circumstances. The authors would like to

express their deepest appreciation to the survey participants and expert panel who contributed to this study. Participants were most generous in sharing their time and experiences.

## Author Contributions

**Conceptualization:** Sanjay Basu, John S. Yudkin, Christopher Millett.

**Data curation:** Mohammed Jawad, Hala Ghattas, Bassam Abu Hamad, Zeina Jamaluddine, Gloria Safadi, Marie-Elizabeth Ragi, Raeda El Sayed Ahmad, Eszter P. Vamos, Christopher Millett.

**Formal analysis:** Sanjay Basu, John S. Yudkin, Mohammed Jawad, Hala Ghattas, Bassam Abu Hamad, Zeina Jamaluddine, Gloria Safadi, Marie-Elizabeth Ragi, Raeda El Sayed Ahmad, Eszter P. Vamos, Christopher Millett.

**Funding acquisition:** Christopher Millett.

**Methodology:** Sanjay Basu, John S. Yudkin, Christopher Millett.

**Supervision:** Sanjay Basu, Christopher Millett.

**Writing – original draft:** Sanjay Basu.

**Writing – review & editing:** John S. Yudkin, Mohammed Jawad, Hala Ghattas, Bassam Abu Hamad, Zeina Jamaluddine, Gloria Safadi, Marie-Elizabeth Ragi, Raeda El Sayed Ahmad, Eszter P. Vamos, Christopher Millett.

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
