## [Decision Letter · Decision Letter 0]

26 Feb 2024

PGPH-D-23-02094

Reducing non-communicable diseases among Palestinian populations in Gaza: a participatory comparative and cost-effectiveness modeling assessment

Dear Dr. Sanjay Basu,

Thank you for submitting your manuscript to PLOS Global Public Health. After careful consideration, we feel that it has merit but does not fully meet PLOS Global Public Health’s publication criteria as it currently stands. Therefore, we invite you to submit a revised version of the manuscript that addresses the points raised during the review process.

Uncertainty analyses:

Inconsistency on what is reported in the methods section and result: confidence interval vs. credible interval. "Uncertainty analyses are reflected in the 95% confidence intervals around the mean results. 

We look forward to receiving your revised manuscript.

Kind regards,

Saifur R. Chowdhury, MPH, PhD (c)

Academic Editor

Journal Requirements:

2. Please provide separate figure files in .tif or .eps format.

Additional Editor Comments (if provided):

Reviewers' comments:

Reviewer's Responses to Questions

**Comments to the Author**

1. Does this manuscript meet PLOS Global Public Health’s publication criteria? Is the manuscript technically sound, and do the data support the conclusions? The manuscript must describe methodologically and ethically rigorous research with conclusions that are appropriately drawn based on the data presented.

Reviewer #1: Yes

Reviewer #2: Yes

Reviewer #3: Yes

2. Has the statistical analysis been performed appropriately and rigorously?

Reviewer #1: Yes

Reviewer #2: Yes

Reviewer #3: Yes

3. Have the authors made all data underlying the findings in their manuscript fully available (please refer to the Data Availability Statement at the start of the manuscript PDF file)?

Reviewer #1: No

Reviewer #2: Yes

Reviewer #3: Yes

4. Is the manuscript presented in an intelligible fashion and written in standard English?

Reviewer #1: Yes

Reviewer #2: Yes

Reviewer #3: Yes

5. Review Comments to the Author

Reviewer #1: I appreciate the opportunity to critically review the manuscript titled "Reducing non-communicable diseases among Palestinian populations in Gaza: a participatory comparative and cost-effectiveness modeling assessment." It is an interesting and highly relevant article that analyzes the health situation of a population with very specific conditions that could have a negative impact on their well-being. While it is true that the study has limitations, appropriately addressed by the authors, I believe the obtained results are highly relevant for public health. I present to the authors a series of minor comments for their consideration.

Comment 1: On page 13, it is stated that more than half of the participants (54%) are women, and almost 7 out of 10 participants identified as refugees. Are these proportions representative of the source population? Please briefly discuss this.

Comment 2: In Table 2 (page 15), I found it concerning that there are no interventions aimed at improving the psychosocial and mental health of this population. I understand that current priorities (January 2024) may differ, but it is an aspect worth briefly discussing.

Comment 3: The political and health situation of the studied population recently changed for reasons beyond the control of the researchers. It would be interesting to briefly discuss the potential impact of these recent events on the burden of the diseases analyzed in this research

Reviewer #2: The author concluded that high levels of NCD risk factors among Palestinians in Gaza have

estimated that several interventions would be expected to reduce the loss of DALYs within

common cost-effectiveness thresholds. So, the author conducted a timely analysis.

Reviewer #3: Given the events that have been happening in Gaza for sometime, it is a perfect timing to have an article of this kind. This can be used as a reference for implementing interventions not only in Gaza but also in context that have been/ are being ravaged by political instabilities. For that, I would like to congratulate the authors.

This article is presented using a simple and clear language, and there is a good coherence in the flow of information. This ensures easy understanding for many readers irrespective of their professional backgrounds.

The method section is detailed and clear. The results and the discussion are well aligned to the objectives and methods used.

6. PLOS authors have the option to publish the peer review history of their article (what does this mean?). If published, this will include your full peer review and any attached files.

**Do you want your identity to be public for this peer review?** For information about this choice, including consent withdrawal, please see our Privacy Policy.

Reviewer #1: No

Reviewer #2: **Yes: **PUGAZHENTHAN THANGARAJU

Reviewer #3: No

---

## [Decision Letter · Decision Letter 1]

8 Apr 2024

Reducing non-communicable diseases among Palestinian populations in Gaza: a participatory comparative and cost-effectiveness modeling assessment

PGPH-D-23-02094R1

Dear Dr. Basu,

We are pleased to inform you that your manuscript 'Reducing non-communicable diseases among Palestinian populations in Gaza: a participatory comparative and cost-effectiveness modeling assessment' has been provisionally accepted for publication in PLOS Global Public Health.

Best regards,

Saifur R. Chowdhury, MPH, PhD (c)

Academic Editor

Reviewer Comments (if any, and for reference):

Reviewer's Responses to Questions

**Comments to the Author**

1. If the authors have adequately addressed your comments raised in a previous round of review and you feel that this manuscript is now acceptable for publication, you may indicate that here to bypass the “Comments to the Author” section, enter your conflict of interest statement in the “Confidential to Editor” section, and submit your "Accept" recommendation.

Reviewer #1: All comments have been addressed

2. Does this manuscript meet PLOS Global Public Health’s publication criteria? Is the manuscript technically sound, and do the data support the conclusions? The manuscript must describe methodologically and ethically rigorous research with conclusions that are appropriately drawn based on the data presented.

Reviewer #1: Yes

3. Has the statistical analysis been performed appropriately and rigorously?

Reviewer #1: Yes

4. Have the authors made all data underlying the findings in their manuscript fully available (please refer to the Data Availability Statement at the start of the manuscript PDF file)?

Reviewer #1: Yes

5. Is the manuscript presented in an intelligible fashion and written in standard English?

Reviewer #1: Yes

6. Review Comments to the Author

Reviewer #1: I am grateful for the opportunity to critically review (Round 1) the manuscript PGPH-D-23-02094R1. The analyzed document is clear and the stated objective is achieved. The comments previously issued by me were duly addressed by the research group and, as I said before, it is a highly relevant research that will be of great interest to readers of PLOS Global Public Health.

7. PLOS authors have the option to publish the peer review history of their article (what does this mean?). If published, this will include your full peer review and any attached files.

**Do you want your identity to be public for this peer review?** For information about this choice, including consent withdrawal, please see our Privacy Policy.

Reviewer #1: **Yes: **Efrén Murillo-Zamora
